# Ultrasensitive and Rapid Detection of N-Terminal Pro-B-Type Natriuretic Peptide (NT-proBNP) Using Fiber Optic Nanogold-Linked Immunosorbent Assay

**DOI:** 10.3390/bios12090746

**Published:** 2022-09-09

**Authors:** Han-Long Liu, Yen-Ta Tseng, Mei-Chu Lai, Lai-Kwan Chau

**Affiliations:** 1Department of Chemistry and Biochemistry, National Chung Cheng University, Chiayi 621301, Taiwan; 2Center for Nano Bio-Detection, National Chung Cheng University, Chiayi 621301, Taiwan; 3Department of Laboratory Medicine, Ditmanson Medical Foundation Chia-Yi Christian Hospital, Chiayi 621301, Taiwan; 4Center for Innovative Research on Aging Society (CIRAS), National Chung Cheng University, Chiayi 621301, Taiwan

**Keywords:** localized surface plasmon resonance, fiber optic nanogold-linked immunosorbent assay, heart failure, N-terminal pro-brain natriuretic peptide

## Abstract

The N-terminal pro-brain natriuretic peptide (NT-proBNP) is considered an important blood biomarker for heart failure. Herein, we report about a fiber optic nanogold-linked immunosorbent assay (FONLISA) method for the rapid, sensitive, and low-cost detection of NT-proBNP. The method is based on a sandwich immunoassay approach that uses two monoclonal NT-proBNP antibodies, a capture antibody (Ab^C^), and a detection antibody (Ab^D^). Ab^D^ is conjugated to a free gold nanoparticle (AuNP) to form the free AuNP@Ab^D^ conjugate, and Ab^C^ is immobilized on an unclad segment of an optical fiber. The detection of analyte (A), in this case NT-proBNP, is based on the signal change due to the formation of an AuNP@Ab^D^–A–Ab^C^ complex on the fiber core surface, where a green light transmitted through the optical fiber will decrease in intensity due to light absorption by AuNPs via the localized surface plasmon resonance effect. This method provides a wide linear dynamic range of 0.50~5000 pg·mL^−1^ and a limit of detection of 0.058 pg·mL^−1^ for NT-proBNP. Finally, the method exhibits good correlation (*r* = 0.979) with the commercial central laboratory-based electrochemiluminescent immunoassay method that uses a Roche Cobas e411 instrument. Hence, our method is potentially a suitable tool for point-of-care testing.

## 1. Introduction

Cardiovascular diseases (CVDs) are the leading cause of death globally, taking an estimated 17.9 million lives each year (32% of all deaths worldwide), according to the World Health Organization [1]. Among those CVDs, acute myocardial infraction (AMI) and congestive heart failure (HF) are the most common. HF is a progressive condition that begins from left ventricular systolic dysfunction (LVSD), proceeds to asymptomatic changes in cardiac structure and function, and then evolves into clinical HF, disability, or even death [2]. Studies show that 25% of people over 85 years old develop HF [3], and that about 6.2 million persons ≥20 years old in the United States have HF, with approximately 1 million new HF cases diagnosed annually [4]. The total cost of HF in the United States exceeds $30 billion annually, with over 50% of spending on hospitalizations; the mortality rate after hospitalization for HF still remains at 20~25% in the first year [4]. Therefore, in order to reduce the burden of CVD in global healthcare, those who are at high cardiovascular risk require early diagnoses and prognoses.

In the current clinical diagnosis of heart failure there are several bases for assessing heart failure. In addition to asking for a patient’s medical history as well as scanning electrocardiograms and chest X-rays, detection of the amount of NT-proBNP in the blood has become an important diagnostic measure. Since B-type natriuretic peptide (BNP) in the porcine brain was described in 1988 [5], diagnosis and prognosis of HF based on the measurement of BNP in blood have been proposed [6]. Although BNP is secreted in the left ventricular (LV) myocardium, the N-terminal pro-B-type natriuretic peptide (NT-proBNP) was found to reflect LV function better than any other neurohumoral factor, since NT-proBNP has longer biological half-life and higher stability in vitro than BNP [7]. Hence, NT-proBNP has been suggested to be a better HF biomarker than BNP [8]. The measurement of BNP or NT-proBNP is useful to monitor risk, to assist in decision-making regarding the ordering of imaging studies to evaluate LV remodeling, and to provide helpful objective data regarding decision-making for referral to advanced HF therapies [9]. The thresholds of serum NT-proBNP levels for the diagnosis of HF are age-dependent, and have been proposed as follows: (1) >1.8 ng/mL for age >75 years, (2) >0.9 ng/mL for age 50–75 years, and (3) <0.45 ng/mL for age <50 years [10].

Thus far, a number of analytical methods in the literature have been reported for the analysis of NT-proBNP, including radioimmunoassay (RIA) [11], colorimetry [12], fluorescent bead-based immunoassay [13], chemiluminescent immunoassay [14], dynamic light scattering [15,16], surface-enhanced Raman scattering [17], amperometric immunosensors based on enzyme label [18], silver nanoparticle label method [19], silver nanodisk label method [20], photoelectrochemical immunosensor method [21], electrochemiluminescent (ECL) immunosensor method [22], field-effect transistor biosensor method [23], giant magnetoresistance assay [24], and fluorescence-based lateral flow immunoassay [25]. While very sensitive electrochemical biosensors for the detection of NT-proBNP have been reported [18,19,20,21,22,23], the requirement of a reference electrode in electrochemical biosensors may be a weak point, as liquid junction error often exists in practical situations [26]. Optical sensors do not require a so-called reference optrode, and also have the advantages of immunity to electromagnetic interference. Several commercial analytical methods have also been reported for the analysis of NT-proBNP, including enzyme-linked immunosorbent assay (ELISA) with a limit of detection (LOD) of 0.12 ng/mL [27], and ECL immunoassay with an analytical range from 5 pg/mL to 35 ng/mL [28]. However, these detection techniques have some disadvantages, such as involving high-cost instrumentation, being time-consuming, requiring a highly skilled operator, and involving unstable isotopes or large sample volumes.

Herein, an assay that employs a fiber optic particle plasmon resonance (FOPPR) sensing platform was developed for the detection of NT-proBNP. The FOPPR biosensing platform is based on the principle of fiber optic evanescent wave absorption by gold nanoparticles (AuNPs) on the unclad fiber core surface. It offers a pronounced signal enhancement advantage through the multiple total internal reflections (TIRs) of light in an optical fiber via the increase in optical path length and field enhancement at the fiber core surface [29,30,31]. As a result, after multiple TIRs, the light transmitted through the fiber is attenuated by interaction with AuNPs. The platform requires just a light-emitting diode (LED), a photodiode (PD), circuit broads to control the system and to acquire data, and an externally connected computer. Thus, it is particularly suitable for point-of-care testing (POCT).

In order to further enhance the analytical sensitivity of the biosensor, a method called fiber optic nanogold-linked immunosorbent assay (FONLISA) was developed recently [32]. The method uses exactly the same instrumentation, but with a new sensing strategy through a unique sandwich architecture. Briefly, as shown in Figure 1, the strategy is implemented by using an immobilized capture antibody (Ab^C^) and a free AuNP-labelled detection antibody (AuNP@Ab^D^) to bind with the target antigen (A); this forms a sandwich-like AuNP@Ab^D^–A–Ab^C^ complex on the fiber core surface. During the binding process, when a green light passes through the fiber sensing region, the transmitted light through the fiber decreases in intensity according to the binding kinetics [33] because AuNP@Ab^D^ is the only light absorption agent; its arrival to the fiber core surface causes absorption of evanescent waves, and hence an attenuation in transmitted light intensity that exits the fiber. By this method, no AuNPs are present on the fiber core surface initially, and thus the background absorbance value is zero. Therefore, the sensitivity of this method depends on what the minimum number of bound AuNPs detectable is; this will set the detection limit for the system. Using the FONLISA approach, quantitative analysis of numerous different species, including procalcitonin [32], ampicillin [34], Hg(II) ion [35], DNA [36], and methylated DNA [37] with ultrahigh sensitivity, have been demonstrated. Here, we describe a rapid, low-cost, and high-sensitivity FONLISA method for the detection of NT-proBNP, with the ultimate aim being for it to be developed as a POCT method.

## 2. Materials and Methods

### 2.1. Chemicals and Materials 

The NT-proBNP (tcup1326, mw. = 12.2 kDa) and sulfobetaine thiol (SB-SH) were obtained from Taiclone (Taipei, Taiwan); the NT-proBNP antibodies 15C4 (IgG2b, a.a.r. 63–71, IgG, immunoglobin G) (a.a.r. = amino acids range), 29D12 (IgG2a, a.a.r. 5–12) were obtained from Hytest (Turku, Finland). Dextran (m.w. ~40 kD, from Leuconostoc spp.), bovine serum albumin (BSA), human serum (product No. H4522), 1-ethyl-3-(3-dimethyl aminopropyl) carbodiimide hydrochloride (EDC·HCl), N-hydroxysuccinimide (NHS), and 16-mercaptohexadecanoic acid (MHDA) were purchased from Sigma-Aldrich (St. Louis, MO, USA). Hydrogen tetrachloroaurate (III) trihydrate was purchased from ThermoFisher Scientific Alfa Aesar (Tewksbury, MA, USA). Ethanolamine and sodium citrate were purchased from ThermoFisher Scientific J.T. Baker (Pittsburgh, PA, USA). Ethanol (99.5%) was obtained from ECHO (Miaoli, Taiwan), and methanol (99.5%) was obtained from Aencore Chemical (Box Hill Vic, Australia). 3-aminopropyl triethoxysilane (APTES 99%) was purchased from ThermoFisher Scientific Acros Organics (Geel, Belgium). Tween-20 was acquired from Showa (Tokyo, Japan). 2-(N-morpholino)ethane sulfonic acid (MES) was obtained from Lancaster University (Lancaster, UK). Carboxymethyl-dextran sodium salt (mw: ~40 kD) was purchased from Tokyo Chemical Industry (Tokyo, Japan). Horseradish peroxidase conjugated goat anti-mouse antibody (HRP-2° antibody, ab6789) was obtained from Abcam (Cambridge, UK). ELISA plate (Nunc MaxiSorp^TM^, 44-2404-21) and the 3,3′,5,5′-tetramethylbenzidine (TMB) substrate kit were purchased from ThermoFisher Scientific (Waltham, MA, USA). PBS tablets were purchased from Abcam BioVision (Waltham, MA, USA). Sucrose came from PanReac AppliChem (Barcelona, Spain). All aqueous solutions and buffers were prepared using pure water obtained from a Milli-Q water system (Merck, Darmstadt, Germany), with a specific resistance of 18.2 MΩ⋅cm. All human plasma specimens were collected under an approved protocol from Ditmanson Medical Foundation, Chia-Yi Christian Hospital Institutional Review Board (IRB number: IRB2018083).

### 2.2. Preparation of Gold Nanoparticles (AuNPs)

The AuNP solution was prepared by the citrate reduction method. A boiled aqueous solution of HAuCl_4_ (0.88 mM, 20 mL) under a reflux system was mixed with an aqueous solution of 1% citrate (2.4 mL). The resulting solution changed color from pale yellow, to colorless, to dark blue, and to ruby red, during the reflux process. The mean size of AuNPs was determined via a transmission electron microscope (TEM, JEOL JEM2010). The AuNP solution was characterized by UV-visible spectroscopy (UV-Vis-NIR, GBC Cintra 2020), with the absorbance peak at 520 nm. When the absorbance of the AuNP solution at 520 nm was set to 1.0 a.u., the concentration of the AuNP solution was estimated to be ~5.0 nM [38].

### 2.3. Conjugation of AuNP with Anti-NT-proBNP Antibody 15C4

The anti-NT-proBNP antibody conjugated AuNP (AuNP@Ab^D^) was prepared as follows: an AuNP solution (~5.0 nM, 2 mL) was mixed with an aqueous solution of Tween-20 (0.2%, 2 mL) at the same volume ratio for 1 h. Then, a 300-microliter mixture of 0.1 mM MHDA and 0.4 mM SB-SH in 75% methanol was added to the above AuNP solution overnight, in order to form a mixed self-assembled monolayer (mSAM) on the AuNP surface. After centrifugation (Hermle Z323K) at 15,700 rcf, the supernatant was discarded and the sediment was re-dispersed with the Tween-20 solution to 4 mL. Then, an aqueous solution of EDC/NHS (13 mM/17 mM, 200 μL) was added and was allowed to react for 30 min. During the reaction, EDC activates the carboxyl groups on the mSAM-modified AuNP, and NHS ester serves as the leaving group. The excess unreacted EDC/NHS was removed by centrifugation at 10,000 rcf under 15 °C for 15 min; then, the sediment was re-dispersed with pure water to 2 mL. After activation, a solution of anti-NT-proBNP 15C4 was added to the solution so that the final antibody concentration was 1.0 × 10^−5^ g·mL^−1^, and the reaction was left to stand overnight. The unreacted NHS esters on the mSAM of the AuNPs were then deactivated by adding a 20-microliter solution of ethanolamine (final concentration = 9.9 × 10^−5^ M). Finally, the AuNP@Ab^D^ conjugate was collected by centrifugation at 10,000 rcf for 20 min, and the sediment was re-dispersed into a solution of 0.2× PBS (NaCl 28.0 mM, KCl 0.5 mM, phosphate buffer 2.0 mM, pH 7.4) in order to obtain the AuNP@Ab^D^ solution.

### 2.4. Characterization of Nanoparticle Size Using DLS

The hydrodynamic diameters of AuNPs and the AuNP@Ab^D^ conjugate were determined by dynamic light scattering (DLS) using a Zetasizer Nano ZS90 Malvern instrument. The measurements were taken at a 90° angle, with 3 repeated measurements. Each measurement comprised 10 runs with a duration of 10 s. Acrylic disposable cuvettes were used for the measurements at room temperature under 25 °C after 120 s of temperature equilibration. For the AuNP solutions used for DLS measurements, the absorbance values at 520 nm of the AuNP solution before and after modification were set at 2.0 a.u. and 0.5 a.u., respectively.

### 2.5. Preparation of Sensing Fibers and Sensing Chips

The optic fibers were multi-mode plastic-clad silica fibers (model F-MBC, Newport, Newport Beach, CA, USA) having core and buffer coating diameters of 400 (±8) and 730 (±30) μm, respectively. A 20 mm sensing region in the central part of an optic fiber was formed by removing the cladding and buffer with a CO_2_ laser processing system (V-460, Universal Laser System Inc., Scottsdale, AZ, USA), in order to expose the core surface. The fiber end faces were polished until optically smooth surfaces for efficient light coupling were achieved.

In order to functionalize the optical fibers with a capture antibody (Ab^C^), firstly the fibers after cleaning were immersed in 4% APTES ethanolic solution for 30 min to undergo silanization before being dried in an oven at 110 °C for 1 h after washing with ethanol and pure water sequentially; this was carried out in order to make the APTES-derived layer flat and even. Secondly, a mixture of CM-dextran/EDC/NHS (0.024/0.015 g/0.030 g) in 3 mL of MES buffer (0.5 mM, pH 6.15) was freshly prepared and reacted with the fiber core surface for amide-bond coupling. Each single modified fiber was placed in a polycarbonate (PC) microfluidic chip that was formed by a pair of cover and bottom plates with dimensions of 2.5 cm in width, 5.0 cm in length, and 0.2 cm in thickness. The bottom plate contained a microchannel with a depth of 900 μm and a width of 900 μm, and the cover plate contained four bored access holes with the inner two holes connected to the microchannel for sample introduction; meanwhile, the outer two holes were for fixing the fiber in place with a glue (epoxy AB glue). A PBS-buffered solution of anti-NT-proBNP antibody 29D12 (1 × 10^−4^ g mL^−1^, 30 μL) was introduced into the sensor chip at 4 °C overnight for anchoring the antibody, and then a solution of 1 M ethanolamine (pH 8.5) was added and allowed to react for 10 min, in order to block the excess activated sites on the CM-dextran layer. After this deactivation step, the sensor chip was washed and conserved with PBS for further use.

### 2.6. Quantitation of Immobilization Antibody on AuNP and Sensing Fiber by ELISA

The average numbers of antibody molecules on each AuNP and sensing fiber were quantified by using anti-mouse IgG HRP secondary antibody based on an ELISA. In order to quantify the immobilized antibody on AuNPs, a standard calibration curve was obtained by preparing serial solutions of 15C4 antibody with concentrations ranging from 0 to 50 ng mL^−1^. Then, 100 μL of each antibody solution was mixed with 100 μL of AuNP solution, and the resulting mixtures were incubated in ELISA plate wells for 18 h. The solutions were pipetted out from wells and washed thrice with 200 μL of TBST solution (20.0 mM Tris, 150 mM NaCl, 0.1% Tween-20, pH 7.4). The unbound sites in wells were blocked with 100 μL of 1.0 μg mL^−1^ BSA for 1.5 h, and washed thrice with 200 μL of TBST. Afterwards, a 100-microliter solution of HRP fused goat-anti-mouse IgG (ab6789) was injected into each well for 1.5 h, and was washed with 200 μL of TBST buffer thrice. After 60 μL of TMB substrate was added to each well and incubated for 15 min, the reaction was stopped by another 60 μL of 2 M sulfuric acid. The absorbance of the ELISA plate was measured at 450 nm under an ELISA reader. In order to measure the amount of 29D12 antibody (Ab^C^) on the fiber core surface, the fibers with immobilized Ab^C^ were immersed in a solution of HRP-fused goat anti-mouse IgG (ab6789) for 1 h, and washed with TBST buffer thrice. Subsequently, the fibers were immersed in a TMB substrate for 15 min. The reaction was stopped with 2 M sulfuric acid; each solution was transferred to the ELISA plate, and the absorbance was measured at 450 nm. The results were performed in triplicate, and the standard deviation was calculated in each case.

### 2.7. Biosensing System 

The biosensing system, as shown in Figure 2, consisted of a 530-nanometer LED (model IF-E93, Industrial fiber optics, Tempe, AZ, USA) as the light source, a sensing module, and a PD (S1336-18BK, Hamamatsu Photonics, Hamamatsu, Japan) as the detector. A self-developed circuit broad comprised an LED driver to drive the LED with 1 kHz frequency modulation. A current amplifier was connected to the PD, a 16-bit analog-to-digital converter (ADC, NI 9215, National Instruments, Austin, TX, USA), a power supply (PMT-DIV100W1AA, Delta Electronics, Inc., Taipei, Taiwan), a computer, and a self-developed graphical interface to demodulate and process the real-time signals by a program in LabVIEW^®^ (National Instruments, Austin, TX, USA). The sensing module comprised a chip holder to place a sensing chip and a sample injector (Rheodyne 7225i, Rheodyne, CA, USA) to load samples into the sensing chip. The output signal from the sensor fiber in PBS was adjusted to about 3 V before each experiment. The relative standard deviation (RSD) of the system’s background noise (σ) was <0.008%, in an average time of 200 s.

In order to accomplish quantitative analyses, the transmitted light intensity from the sensing chip where the microchannel was filled with a blank (I_0_) was compared to that with a sample (I_x_), where I_x_ is the steady-state intensity calculated as an average of 100 steady-state data points. Since the principle of FOPPR measurements is based on absorption spectroscopy, we defined the normalized intensity (I_0_ − I_x_)/I_0_ = ΔI/I_0_ as the sensor response which allowed for a certain degree of variation in I_0_ for different sensing chips; hence, precise alignment of every sensing chip to obtain the same I_0_ was not required.

### 2.8. Preparation of Standards 

In order to construct a calibration curve for NT-proBNP detection, a serial dilution of the stock solution of NT-proBNP with PBS was performed to yield standard solutions with NT-proBNP concentrations ranging from 1 pg·mL^−1^ to 100,000 pg·mL^−1^. Then, 200 μL of each standard solution was mixed with 200 μL of AuNP@Ab^D^ solution (0.4 a.u., 1.62 nM) and incubated for 15 min to yield solutions with final NT-proBNP concentrations ranging from 5 × 10^−13^ – 5 × 10^−9^ g·mL^−1^, which were injected into the sensing chip sequentially, from low to high concentrations, with each injection lasting 15 min to reach the steady-state signal. The steady-state signals of every concentration were then used to calculate the sensor responses. A plot of sensor responses with log concentrations of NT-proBNP was employed as the standard calibration curve.

### 2.9. Preparation and Measurements of Clinical Specimens

The plasma specimens from patients were frozen at −20 °C after collection at Chia-Yi Christian Hospital. The NT-proBNP concentrations of the samples were determined using a clinical standard ECLIA method (Roche Elecsys NT-proBNP assay, Cobas e411 analyzer), at a similar time to when the FONLISA method was performed. Before FONLISA measurements, the thawed plasma samples were diluted by PBS at a 1:1 volume ratio. After 20 min of centrifugation at 18,000 rcf, 100 μL of the supernatant was diluted with 900 μL of PBS buffer. From this diluted solution, 100 μL was taken out to be further diluted with 900 μL of PBS buffer; then, 200 μL of the final diluted solution was mixed with 200 μL of AuNP-Ab^D^ solution (0.4 a.u., 1.62 nM) so that the overall dilution factor was 800. The resulting mixtures were incubated for 15 min, and then injected into the sensing chip, waiting 15 min to allow the reaction to reach equilibrium. In order to study the matrix effect and to estimate recovery, known concentrations of NT-proBNP were spiked into human serum samples and diluted with PBS buffer to reach a dilution factor of 20. Then, the same procedures as for the real samples, including the mixing with AuNP@Ab^D^ solution and injection into the sensing chip, were followed.

## 3. Results and Discussion

### 3.1. Characterization of AuNP and AuNP@Ab^D^ Conjugate

The citrate-capped gold nanoparticles were synthesized using the citrate reduction method and characterized by UV-vis spectroscopy, TEM, and DLS. As shown by the TEM image in Figure 3C, the mean particle size of AuNPs was 13.9 ± 1.1 nm. By UV-vis spectroscopy, as shown in curve a of Figure 3B, the AuNP solution showed a peak wavelength at 520 nm, a full width at the half maximum of 84 nm, and a peak absorbance of about 2.1 a.u. due to the particle plasmon resonance of AuNPs. This AuNP solution was further diluted by 0.1% sodium citrate solution until the absorbance was 1.0 a.u., and used as the stock solution. The particle concentration of the stock AuNP solution was about 5.0 nM, estimated by its relationship with the gold atom concentration and particle statistical mean volume [38].

In Figure 3A(a) citrate-capped AuNPs were first modified with a mSAM that was formed by chemisorption of MHDA and SB-SH, and denoted as AuNP−SAM (Figure 3A(b)). After the formation of the mSAM on the AuNP surface was completed, NT-proBNP antibody 15C4 was subsequently conjugated to the AuNP−SAM via the EDC/NHS chemistry (Figure 3A(c)). As such, the peak wavelength in the absorption spectrum of AuNP solution red-shifted to 523 nm after modification with the mSAM (Figure 3B, curve b), and then to 527 nm after antibody conjugation (Figure 3B, curve c), due to the increase in local refractive index (RI) of the surrounding medium around the AuNPs [39,40]. As there were some losses of AuNPs during each step of the process, UV-vis spectroscopy was also utilized to estimate the overall loss; about 52.8% loss was found. Hence, the concentration of AuNP@Ab^D^ in the final solution was estimated to be 2.5 nM.

DLS was further used to verify the success in the steps of surface modification by mSAM and conjugation of detection antibody on AuNPs. As shown in curve a of Figure 3D, the mean hydrodynamic diameter of citrate-capped AuNPs was 16.2 ± 5.0 nm. Upon modification with the mSAM, the mean hydrodynamic diameter of the AuNP–SAM was estimated to be 23.7 ± 6.2 nm (Figure 3D, curve b). Upon further conjugation with the detection antibody, the mean hydrodynamic diameter of AuNP@Ab^D^ was estimated to be 61.3 ± 24.1 nm (Figure 3D, curve c).

### 3.2. Quantitation of Antibody Immobilized on AuNPs and Fiber

The number of antibody molecules that can be immobilized on each AuNP and on the fiber-sensing region mainly depends on the method of immobilization used. Previously, we developed a method based on ELISA to quantify the number of antibody molecules on the optical fiber surface [34]. This study employed the same method to quantify the number of antibody molecules on each fiber, and also extended this approach to the quantitation of the average number of antibody molecules on each AuNP. The quantification was determined from the absorbance increase at 450 nm upon enzymatic oxidation of TMB by HRP, which could be interfered with by the optical absorption of AuNP at 450 nm. Therefore, baseline correction using an AuNP solution of the same concentration but with zero concentration of detection antibody was employed. The molecular weight of an antibody molecule is about 150 kD. In order to determine the average number of antibody molecules on each AuNP, after aliquots of 100 μL of 0.05 a.u. AuNP solution (~2.5 × 10^−10^ M) were mixed with various amounts of the detection antibody, a calibration curve following the method described in Section 2.6 was constructed. As shown in Figure 4a, the quantity of detection antibody in the AuNP@Ab^D^ solution for the biosensing experiments was found to be 4.00 ng, which is equivalent to 1.61 × 10^10^ molecules. Since the number of AuNPs was estimated to be about 1.51 × 10^10^, therefore the average number of antibody molecules per AuNP is about 1.07. Considering the sizes of an AuNP and an antibody molecule are very similar, it is reasonable to see that essentially one antibody molecule is conjugated to each AuNP.

In order to determine the number of antibody molecules on the fiber-sensing region, the method as previously described was followed [34]. Using the calibration as shown in Figure 4b, the quantity of capture antibody on one optical fiber was estimated to be 8.81 ng, which is equivalent to 3.54 × 10^10^ molecules. Assuming that the size of an IgG molecule is 13.9 nm in length, 12.2 nm in height, and 4.0 nm in thickness [41], and that the IgG molecule with the Fc region stands on the surface, the projected area is 0.0139 μm × 0.004 μm = 5.6 × 10^−5^ μm^2^. In other words, the maximum number of antibody molecules to form a close-packed monolayer in 1 μm^2^ is 1.8 × 10^4^ molecules/μm^2^. Since the area of the sensing region was 2.51 × 10^7^ μm^2^, the number of antibody molecules on the fiber was calculated to be 1.41 × 10^3^ molecules/μm^2^, which is equivalent to a surface coverage of about 7.8%.

### 3.3. Background Nonspecific Adsorption

Nonspecific adsorption (NSA) of AuNP@Ab^D^ conjugate on the sensing fiber surface will cause a false positive signal, while NSA of AuNP@Ab^D^ conjugate among themselves will complicate the sensing results. Therefore, improving the anti-NSA property of both the AuNP@Ab^D^ conjugate and sensing fiber surface is an important issue in enhancing the signal-to-noise ratio of the biosensor. In this study, we used a dextran coating on the fiber core surface, and an mSAM made of the sulfobetaine (SB) group and −COOH group to improve the anti-NSA property of both surfaces. Dextrans are non-charged hydrophilic natural polymeric carbohydrates that show very low nonspecific interactions with proteins [35,36,37,42]. In the mSAM, the SB group is a zwitterionic functional group which has been demonstrated to have a superb anti-NSA property [32,35,36,37,43], while the −COOH group supplies a conjugation site for the antibody.

The background NSA test was performed via injection of a solution of AuNP@Ab^D^ (~2.5 nM) without NT-proBNP into a Ab^C^-functionalized sensing chip, as shown in Figure 5. The average sensor response (ΔI/I_0_) was 0.00177 ± 0.00046 (*n* = 5). As background NSA of AuNP@Ab^D^ onto the fiber-sensing region is considered background chemical noise, a maximum background adsorption level (BAL) is defined here as the mean plus three times the standard deviation of the background NSA of AuNP@Ab^D^ on the fiber-sensing region. Therefore, the BAL of this assay was calculated to be 0.00315. In other words, only the sensor response higher than the BAL can be considered as a positive signal.

### 3.4. Calibration Curve and Study of Matrix Effect

The FONLISA method employs AuNP as a signal reporter to label the detection antibody Ab^D^, in order to construct a calibration curve. In this study, standard NT-proBNP samples were incubated with a fixed concentration of AuNP@Ab^D^ (~2.5 nM) for 15 min before injection into a sensing chip. Upon sequential injection of the incubated samples, from low to high concentrations of NT-pro-BNP, the amount of AuNP@Ab^D^–A–Ab^C^ complex formed on the fiber sensing region increased accordingly, resulting in a typical sensorgram as shown in Figure 6a. It can be seen that for each sample injection, the real-time signal follows the molecular binding kinetics between an immobilized receptor and a free ligand, and finally reaches the steady-state at equilibrium [33]. At the end of the sequential injections, a PBS solution was further injected to observe the stability of the bound AuNP@Ab^D^ conjugate on the fiber core surface. It can be seen that the signal remains essentially unchanged, suggesting that the bound AuNP@Ab^D^ conjugate was not desorbed by PBS. The linear dynamic range of the calibration curve, as shown in Figure 6b, is from 0.50 to 5000 pg·mL^−1^. As the clinical NT-proBNP cut-off values are age-dependent from 450 to 1800 pg·mL^−1^, the wide dynamic range (five orders) of this calibration curve covers the range of all cut-off values. The linear regression equation obtained from the calibration curve is *y* = 0.0065*x* + 0.0112, where *x* = log concentration of NT-proBNP, and the correlation coefficient *r* = 0.999. The limit of detection (LOD) was calculated to be 0.058 pg·mL^−1^ (4.8 fM), assuming BAL as the minimum distinguishable analytical signal. The limit of quantitation (LOQ) was calculated to be 0.181 pg·mL^−1^ (14.8 fM), assuming that the minimum reliably quantifiable signal is equal to the mean plus 10 times the standard deviation of the background NSA of AuNP@Ab^D^ on the fiber-sensing region. The method is highly reproducible, with coefficient of variation (CV) between 2.1 to 6.5% over the whole concentration range.

Selectivity is an important issue for developing a new analytical method. The dual-binding site sandwich format immunoassay is generally more specific than its single-binding site format counterparts, including direct assay and competitive assay, because the target protein is sandwiched between two highly specific antibodies [44,45]. In order to evaluate the matrix effect on the sensor response and the specificity of the biosensor, human serum was spiked into three standard samples with NT-proBNP concentrations of 1.0 × 10^−12^, 1.0 × 10^−10^, and 2.5 × 10^−10^ g·mL^−1^. They were then mixed with an AuNP@Ab^D^ solution (~2.5 nM) in a 1:1 volume ratio, in order to compare the sensor responses. By this procedure, the dilution factor for serum was 20. As shown in Figure 6b and Table 1, the sensor responses with all three serum-spiked NT-proBNP standards fell onto the calibration line well, with CV values between 0.9% and 5.1%, and the recoveries between 88.6% and 104.9%. This suggests that the anti-NSA layers on both the fiber-sensing region and AuNP surface are effective, even at a dilution factor of 20 for clinical samples, and that the biosensor is selective in human serum matrix.

### 3.5. Method Validation with Clinical Plasma Specimens

In order to validate the accuracy of the FONLISA method for NT-proBNP, 13 plasma specimens provided by Ditmanson Medical Foundation, Chia-Yi Christian Hospital, were evaluated by using FONLISA and Roche Cobas e411 separately in a double-blinded manner. The plot of the results from FONLISA versus that from Roche Cobas e411 shows a good correlation coefficient (*r* = 0.979) and a *p* value < 0.0001 at the 95% confidence intervals from the linear regression analysis, as shown in Figure 7. The regression line of the plot has a slope of 1.049 and an intercept −0.741, suggesting that the FONLISA method has small proportional bias and constant bias, respectively, as compared to the clinically accepted electrochemiluminescent immunoassay method.

### 3.6. Comparison with Other Optical Biosensors

Recently, a wide range of optical biosensing techniques have been developed for determination of NT-proBNP, including colorimetry [12], fluorescence [13], chemiluminescent immunoassay [14], dynamic light scattering (DLS) [15], surface-enhanced Raman scattering (SERS) [16], and fluorescent immunochromatographic assay (F-ICA) [25]. Herein, we present a method of detecting and quantifying NT-proBNP concentrations via an immunoassay that employs an antibody sandwich model. The aim of this research was to utilize the ultra-sensitivity of the FONLISA to develop a rapid and low-cost biosensor for a POCT application. As shown in Table 1, although the DLS and SERS methods are more sensitive than our FONLISA method, they require expensive analytical instrumentation and well-trained personnel. On the other hand, our FONLISA has a short analysis time but a better LOD as compared to other affordable optical methods.

## 4. Conclusions

This research illustrates a new approach that utilizes AuNP as a reporter to detect NT-proBNP with ultra-high sensitivity (LOD = 0.058 pg·mL^−1^) and short analysis times (≤15 min). At a dilution factor of 20 for serum samples, the good recoveries between 88.6% and 104.9% suggest that a matrix effect is negligible beyond the dilution factor of 20 for real samples. The LOD and LOQ of this method are very low, so that even at a dilution factor of 800 for clinical samples, the sensitivity still reaches all cut-off values for the diagnosis of HF. Furthermore, the wide linear dynamic range (five orders) of this method covers the range of all cut-off values and high-concentration clinical samples for the diagnosis of HF. Moreover, the method is highly reproducible, with CV values between 2.1 to 6.5% over the entire concentration range. As compared to the commercial electrochemiluminescent immunoassay method, Roche Cobas e411, a gold standard in NT-proBNP immunoassay test, the results by both methods are well-correlated. This suggests that the FONLISA method can be an alternative to the central laboratory-based electrochemiluminescent immunoassay method. In summary, the FOPPR biosensor using the FONLISA method provides a rapid, ultra-sensitive detection approach for NT-proBNP, with a wide linear dynamic range at a low cost, suggesting that the FONLISA is a potentially suitable tool for POCT.

## Figures and Tables

**Figure 1 biosensors-12-00746-f001:**
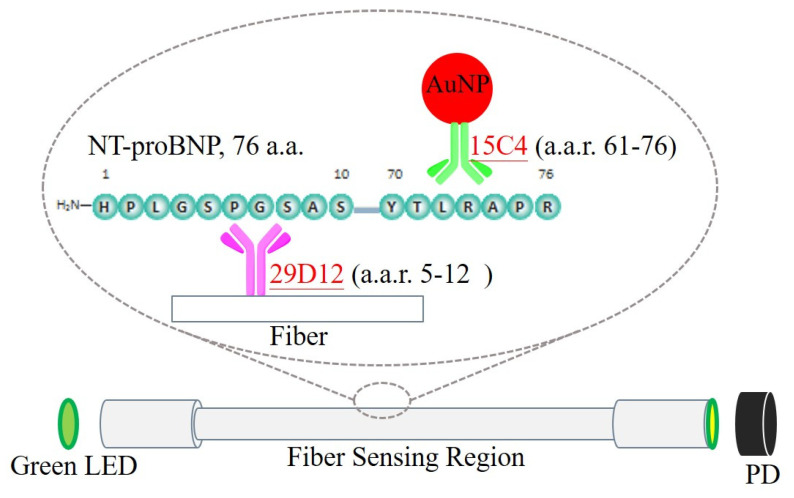
Schematic representation of the FONLISA strategy in the FOPPR sensing system for detection of NT-proBNP. The capture antibody (Ab^C^, 29D12) is immobilized on the sensing region, and the detection antibody (Ab^D^, 15C4) is conjugated to AuNP. When an analyte is sandwiched between the two antibodies, the green light generated from the LED (532 nm) will pass beneath the sensing region and be absorbed by AuNP to finally reach the PD.

**Figure 2 biosensors-12-00746-f002:**
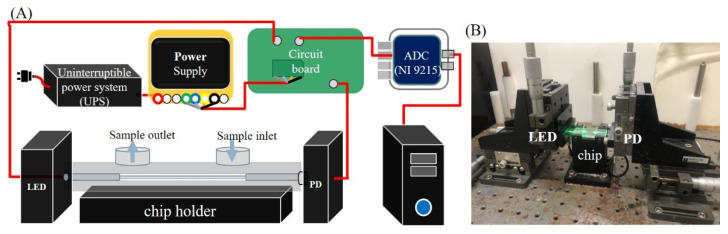
(**A**) Schematic illustration of the experimental setup used for the biosensing system. (**B**) A photograph of the setup.

**Figure 3 biosensors-12-00746-f003:**
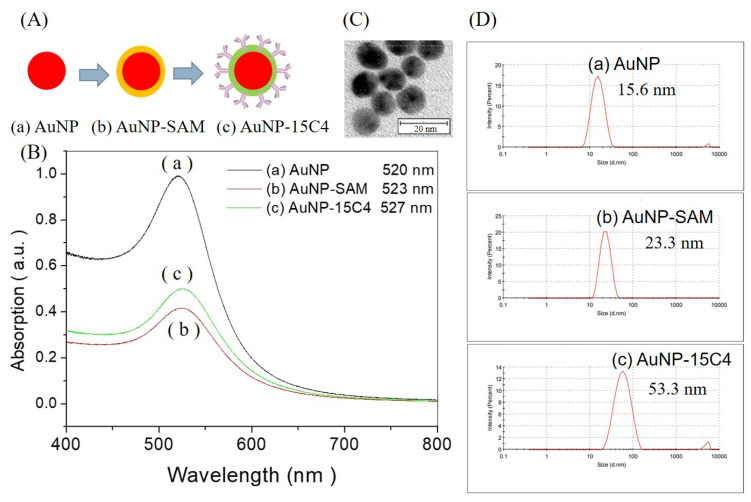
(**A**) Schematic of the process from citrate-capped AuNP to AuNP-SAM, and then to AuNP@Ab^D^ conjugate. (**B**) Corresponding UV-vis spectra. (**C**) TEM image of citrate-capped AuNPs. (**D**) Corresponding representative DLS results.

**Figure 4 biosensors-12-00746-f004:**
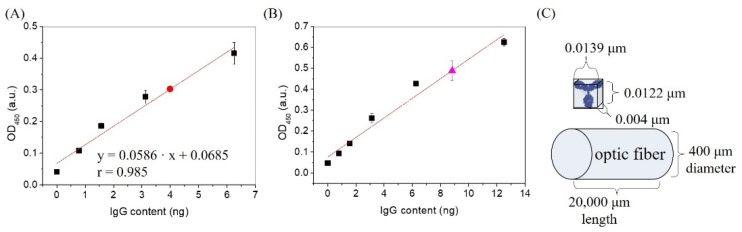
Calibration curves for quantitation of detection antibody and capture antibody immobilized on (**A**) AuNP and (**B**) sensing fiber, respectively. The black square data points represent the calibration points. The red circle and magenta triangle data points show the values obtained for the amount of antibody molecules immobilized on (**A**) AuNP and (**B**) sensing fiber, respectively. (**C**) Size estimation of an antibody.

**Figure 5 biosensors-12-00746-f005:**
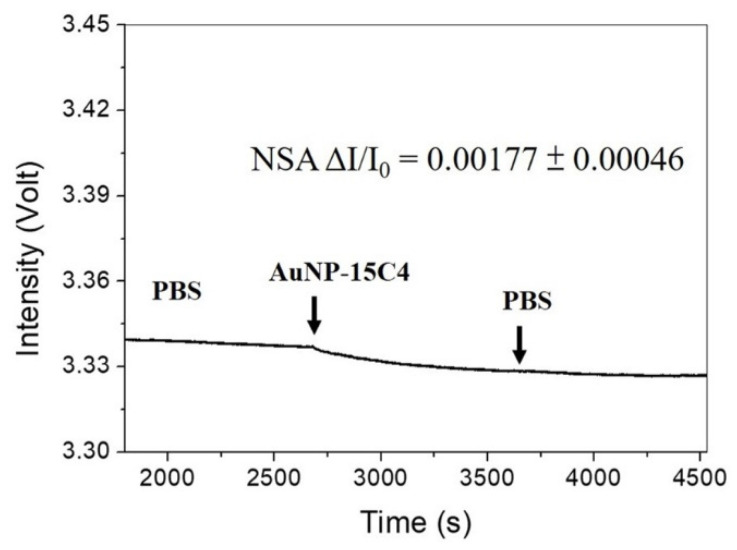
Real-time sensorgram showing the NSA responses of a sensing fiber upon injection of an AuNP@Ab^D^ solution (~2.5 nM), and then a PBS solution.

**Figure 6 biosensors-12-00746-f006:**
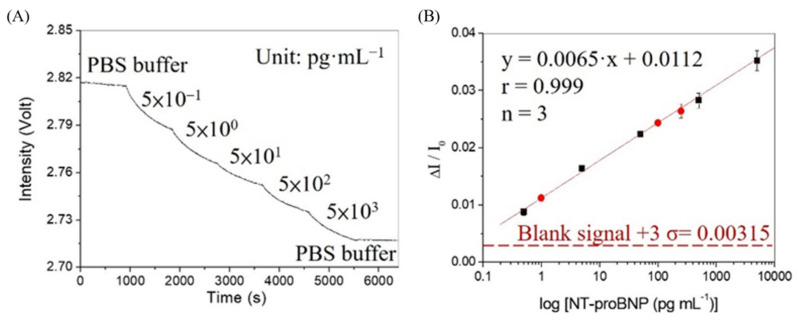
(**A**) Sensorgram in response to sequential multiple injections of NT-proBNP standards with concentrations of 0.5, 5.0, 50, 500, and 5000 pg mL^−1^. (**B**) Corresponding standard calibration curve for NT-proBNP, with all calibration points denoted as black squares (*n* = 3). Sensor responses with serum-spiked NT-proBNP standards at concentrations of 1.0 × 10^−12^, 1.0 × 10^−10^, and 2.5 × 10^−10^ g mL^−1^ are denoted as red circles (*n* = 3). Red dashed line represents the BAL.

**Figure 7 biosensors-12-00746-f007:**
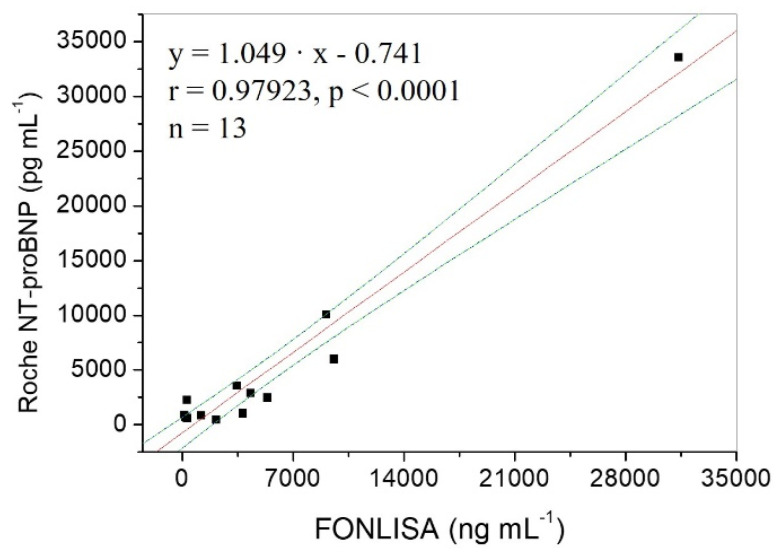
Correlation of results between FONLISA and Roche Cobas e411 for detection of NT-proBNP in blood plasma samples. The red line is the regression line, and the blue dashed lines are the 95% confidence intervals for the regression model.

**Table 1 biosensors-12-00746-t001:** Comparison of optical biosensors for NT-proBNP detection.

Methodology	Linear Range (pg/mL)	LOD (pg/mL)	Time (min)	Reference
Colorimetry	100–100,000,000	70	−	[12]
Fluorescence	7−600	3.7	30	[13]
Chemiluminescence	5.7−6450	0.58	25	[14]
DLS	0.012−100	0.0074	20	[15]
SERS	0.001−1000	0.0075	−	[16]
F-ICA	200−26,000	47	15	[25]
FONLISA	0.50−5000	0.058	15	This work

## Data Availability

The data presented in this study are available in this article here.

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
