# Peer review of "Ultrasensitive and Rapid Detection of N-Terminal Pro-B-Type Natriuretic Peptide (NT-proBNP) Using Fiber Optic Nanogold-Linked Immunosorbent Assay"

_biosensors, 2022, doi:10.3390/bios12090746_

Round 1

Reviewer 1 Report

Manuscript «Ultrasensitive and Rapid Detection of N-terminal Pro-B-type 3 Natriuretic Peptide (NT-proBNP) Using Fiber Optic Nanogold4 Linked Immunosorbent Assay» is very well written. The topic covers the important topic of sensors for determining pro-BNP. The experimental part and the results are described in detail. Limit of detection is incredible.

Comments

1) Line 153 MHDA is used in different way. Most of place 16 - MHDA;

2) it would be great to describe in more detail similar systems based on gold nanoparticles. So your contribution will be visible to a greater extent;

3) It would be nice to insert a photo of the installation. It will be clearer how the sensor is attached.

Author Response

General Comment. Manuscript «Ultrasensitive and Rapid Detection of N-terminal Pro-B-type 3Natriuretic Peptide (NT-proBNP) Using Fiber Optic Nanogold4 Linked Immunosorbent Assay» is very well written. The topic covers the important topic of sensors for determining pro-BNP. The experimental part and the results are described in detail. Limit of detection is incredible.

Response: Thank you very much for your support.

Comments

Comment 1. Line 153 MHDA is used in different way. Most of place 16 - MHDA;

Response: Thank you for pointing out our inconsistency in the use of the term! We now unify the terms as MHDA.

Comment 2. it would be great to describe in more detail similar systems based on gold nanoparticles. So your contribution will be visible to a greater extent;

Response: Thank you for your suggestion! We add more descriptions and references about our similar systems based on gold nanoparticles in the last paragraph of the Introduction section as “Using the FONLISA approach, quantitative analysis of numerous different species including procalcitonin [32], ampicillin [34], Hg(II) ion [35], DNA [36], and methylated DNA [37] with ultrahigh sensitivity have been demonstrated.”.

Comment 3. It would be nice to insert a photo of the installation. It will be clearer how the sensor is attached.

Response: Thank you for your suggestion! We add a new figure (now Figure 2) in Section 2.7 “Biosensing system”, which shows the schematic layout of our biosensing system and a photo of the installation.

Author Response

General Comment. This manuscript reports on the N-terminal pro-brain natriuretic peptide (NT-proBNP), which is an important blood biomarker, considered responsible for the heart failure: (i) Fiber optic nanogold-linked immunosorbent assay (FONLISA) method employed for the rapid, sensitive, and low-cost detection of NT-proBNP, providing a wide linear dynamic range of 0.50 ~ 5000 pg·mL−1 and a limit of detection of 0.058 pg·mL−1 for NT-proBNP. (ii) The FONLISA method accuracy is validated for NT-proBNP, 13 plasma specimens provided by Ditmanson Medical Hospital by using FONLISA and Roche Cobas e411 separately in a double-blinded manner. Overall, the work has merit and results obtained are acceptable. This manuscript can be considered for publication after optional minor revisions.

Response: Thank you very much for your support.

Comment 1. The authors should cite more recent articles on the detection of NT-proBNP.

Response: Thank you for your comment! We add some other new biosensors for NT-proBNP with three more recent articles published between 2021 and 2022 in the Introduction section, 3rd paragraph.

Comment 2. The authors have presented the problem perspective in detail: Starting from the disease statistics in the world, a thorough research background, current status, problems and their solution.

P2lines 77-79 – It is mentioned that “several commercial analytical methods have also been reported for the analysis of NT-proBNP, including enzyme-linked immunosorbent assay (ELISA), etc.” Can the authors provide a comparison with other techniques so that the utilized method can be presented in most distinctive way?

Response: We add more information about the commercial analytical methods in order to compare with our method. Comparison with other optical biosensors had also been done in the Section 3.6 and Table 1. Thank you!

Comment 3. P5Line227 – The modulation frequency in kilohertz range is written “k-Hz”, I think it will be more appropriate to remove the dash. This unit is mostly denoted without the insertion of dash.

Response: Sorry for the typos. We correct it as kHz. Thank you!

Comment 4. P7lines297-300 – Mean hydrodynamic diameters mentioned in the sentence “Upon modification with a mSAM, the mean hydrodynamic diameter of AuNP−SAM was estimated to be 23.7 ± 6.2 nm (Figure 2D, Curve b)” have contradicting Significant Figures in the Fig. 2D and in the text. For example, in the text 23.7 nm with 3 Significant Figurers different from mentioned value 23 in the text having 2 Significant Figures.

Response: We unify the DLA results to 3 Significant Figurers in both text and figure. Thank you!

Comment 5. P8lines344-346 – The sentence in these lines “Therefore, how to improve the anti-NSA property of both the AuNP@AbD conjugate and sensing fiber surface is an important issue to enhance the signal-to-noise ratio of the biosensor” could be rewritten for improved clarity “Therefore, improving the anti-NSA property of both the AuNP@AbD conjugate and sensing fiber surface is an important issue in enhancing the signal-to-noise ratio of the biosensor.”.

Response: We revise it according to your suggestion. Thank you!

Reviewer 3 Report

The manuscript reports an assay using fiber optic particle plasmon resonance for the low-cost, rapid, and sensitive detection of N-terminal pro-brain natriuretic peptide. The method used an interesting sandwich immunoassay approach of two monoclonal antibodies and gold nanoparticles, when reacted the signal could change due to the formation of AuNP@AbD-A-AbC complex and the detection limit is as low as 0.058 pg/mL in a wide linear dynamic range of 0.50 ~ 5000 pg/mL. Overall, this work is interesting and I have no serious criticisms regarding the methodology and results, except for a few questions listed below, and the manuscript can be accepted for publication after the revision.

1. What is the stability of the conjugated gold nanoparticles on the fiber?

2. Selectivity is an important issue for developing a new analytical method and how is the selectivity of the method.

Author Response

General Comment. The manuscript reports an assay using fiber optic particle plasmon resonance for the low-cost, rapid, and sensitive detection of N-terminal pro-brain natriuretic peptide. The method used an interesting sandwich immunoassay approach of two monoclonal antibodies and gold nanoparticles, when reacted the signal could change due to the formation of AuNP@AbD-A-AbC complex and the detection limit is as low as 0.058 pg/mL in a wide linear dynamic range of 0.50 ~ 5000 pg/mL. Overall, this work is interesting and I have no serious criticisms regarding the methodology and results, except for a few questions listed below, and the manuscript can be accepted for publication after the revision.

Response: Thank you very much for your support.

Comment 1. What is the stability of the conjugated gold nanoparticles on the fiber?

Response: Thank you for raising up this important question! The stability of the conjugated gold nanoparticles on the fiber had been demonstrated in Figure 5 (now Figure 6) by injection of a PBS solution after the complex product was formed on the fiber. We add more description of the experiment in the revised Section 3.4 as “At the end of the sequential injections, a PBS solution was further injected to observe the stability of the bound AuNP@AbD conjugate on the fiber core surface. It can be seen that the signal remains essentially unchanged, suggesting that the bound AuNP@AbD conjugate was not desorbed by PBS.”

Comment 2. Selectivity is an important issue for developing a new analytical method and how is the selectivity of the method.

Response: Thank you for raising up this important question! We add more descriptions about the selectivity of the method in the revised Section 3.4 as “The dual-binding site sandwich format immunoassay generally is more specific than its single-binding site format counterparts including direct assay and competitive assay because the target protein is sandwiched between two highly specific antibodies [44,45]. To evaluate …… and the specificity of the biosensor, …… and the biosensor is selective in human serum matrix”.